# SENTINEL: MULTI-PATCH TRANSFORMER WITH TEMPORAL AND CHANNEL ATTENTION FOR TIME SERIES FORECASTING

## ABSTRACT

Transformer-based time series forecasting has recently gained strong interest due to the ability of transformers to model sequential data. Most of the state-of-the-art architectures exploit either temporal or inter-channel dependencies, limiting their effectiveness in multivariate time-series forecasting where both types of dependencies are crucial. We propose Sentinel, a full transformer-based architecture composed of an encoder able to extract contextual information from the channel dimension, and a decoder designed to capture causal relations and dependencies across the temporal dimension. Additionally, we introduce a multi-patch attention mechanism, which leverages the patching process to structure the input sequence in a way that can be naturally integrated into the transformer architecture, replacing the multi-head splitting process. Extensive experiments on standard benchmarks demonstrate that Sentinel, because of its ability to "monitor" both the temporal and the inter-channel dimension, achieves better or comparable performance with respect to state-of-the-art approaches.

## 1 INTRODUCTION

Time-series forecasting is a critical task in many real-world applications, hence the interest in researching better techniques in this field is high. Recent advancements in deep learning have provided new approaches to tackle this problem. However, effectively modeling multivariate time series, where complex relationships exist between multiple features, remains a challenging task. In this perspective, capturing both long-term dependencies and inter-channel relationships is crucial for improving forecast accuracy.

Transformers (Vaswani et al., 2017) have shown promise in handling long-range dependencies in various domains, including natural language processing (Brown et al., 2020) and computer vision (Dosovitskiy et al., 2021). Some recent works have extended transformers to the time-series forecasting task, with mixed results. While some studies demonstrate their effectiveness in capturing temporal relationships(Wang et al., 2024; Liu et al., 2024; Zhang & Yan, 2023), others highlight limitations in addressing inter-channel dependencies efficiently (Nie et al., 2023; Jin et al., 2024; Chang et al., 2023).

The current literature on transformers for time-series forecasting can be mainly divided in two trends: developing new transformer architectures (Liu et al., 2024; Wang et al., 2024; Nie et al., 2023; Zhang & Yan, 2023) or fine-tuning pre-trained models (Zhou et al., 2023; Chang et al., 2023). While both directions are promising, in this work we focus on the first approach where we identify a gap in the current literature: most transformer-based approaches focus on either temporal or inter-channel dependencies but only few works try to model both these dependencies (Wang et al., 2024). This limits the ability of state-of-the-art models to fully exploit the intricate relationships within multivariate time-series data.

Against this background, we designed a full transformer-based architecture capable of capturing both temporal and inter-channel relationships. The encoder is responsible for modeling inter-channel dependencies, learning how different features (or channels) in the time series relate to each other on a patch basis across the entire sequence. This allows the encoder to generates a global context for each channel.

On the other hand, the decoder specializes in capturing temporal dependencies searching for causal relationships across the time dimension. The input to the decoder first passes through a self-attention layer, and the output of this layer serves as the query for the cross-attention mechanism. To further enhance prediction accuracy, the cross-attention mechanism integrates the channel-wise context generated by the encoder with the temporal query produced by the decoder in the previous step, enabling the model to effectively forecast future time steps.

Additionally, the division of the time series into patches (Nie et al., 2023) has shown to be helpful in the forecasting task: it simplifies the discovery of closer relations between patches and reduces the computational complexity of the model.

Following this idea we propose a modified attention mechanism. It takes advantage of the patching process, replacing the traditional multi-head attention with a multi-patch attention, to shift the focus from the concept of "head" to that of "patch". This multi-patch mechanism is utilized throughout the entire architecture: in the encoder, where each "head" independently attends to the sequence at patch level, extracting the channels relationships between the various time steps, and in the decoder, where each "head" independently attends to the sequence at patch level, extracting the temporal relationships between the various time steps.

Summarizing, our key contributions are:

- A novel multi-patch attention mechanism able to exploit the patched input, redesigning the multi-head attention.
- A specialized encoder-decoder architecture that is able to capture both temporal and inter-channel dependencies, where the encoder focuses on inter-channel relationships and the decoder captures temporal dependencies, enhancing the model's forecasting capabilities.
- An extensive evaluation on standard benchmarks for time-series forecasting, showing that the proposed architecture achieves better or comparable performances with respect to other state-of-the art approaches.
- An ablation study demonstrating that the components we propose contribute significantly to enhance overall forecasting performance.

## 2 RELATED WORK

Several transformer-based architectures have proven to be effective in time series forecasting, leading to the development of various methods aimed at improving performance and reducing computational complexity.

The key ingredient that allows transformer-based architectures to achieve good result in time series forecasting is the attention mechanism. Several works have focused on reducing computational complexity and memory usage. For instance, LogTrans (Li et al., 2019) introduces a convolutional self-attention mechanism alongside a LogSparse Transformer to reduce the complexity of standard attention mechanism. Informer (Zhou et al., 2021) proposed a ProbSparse self-attention mechanism to address memory usage and computational overhead. In contrast, Autoformer (Wu et al., 2021) proposes an auto-correlation mechanism. FedFormer (Zhou et al., 2022) enhances transformer architectures with frequency domain information. Pyraformer (Liu et al., 2022) develops a pyramidal attention module able to summarize features at different resolutions. Other works, such as SAM-former (Ilbert et al., 2024) aim to limit overfitting by proposing a shallow lightweight transformer model to escape local minima.

PatchTST (Nie et al., 2023) introduces the concept of patch and channel independence, which has led to significant improvements in forecasting performance. In our work, we exploit the patch structure created through their idea, but we leverage it by considering the full channel relationships. Additionally, other works have attempted to challenge the notion of channel independence, demonstrating that exploring relationships at the channel level is crucial. Crossformer (Zhang & Yan, 2023) implements a Two-Stage Attention mechanism designed to capture both cross-time and cross-dimension dependencies. iTransformer (Liu et al., 2024) applies the attention and feed-forward network on the temporal dimension to improve forecasting performance. CARD (Wang et al., 2024) introduces a channel-aligned transformer combined with a signal decay-based loss function. With respect to CARD, which implements an encoder-only structure to model both temporal and cross-channels

dependencies, we propose an encoder-decoder architecture that specializes the encoder in capturing contextual information through the channel dimension while allowing the decoder to focus on modeling causal relations and dependencies across the temporal dimension.

Another trend focuses on leveraging pre-trained models for time-series forecasting. LLMTime (Nate Gruver & Wilson, 2023) presents a zero-shot approach by encoding numbers as text. lag-llama (Rasul et al., 2024) proposes a foundation model for univariate probabilistic time series forecasting. GPT4TS (Zhou et al., 2023) demonstrates the universality of transformer models and emphasizes the importance of freezing attention layers during fine-tuning. Time-LLM (Jin et al., 2024) introduces a reprogramming framework to repurpose LLMs for general time series forecasting tasks. LLM4TS (Chang et al., 2023) employs a two-stage fine-tuning strategy for utilizing pre-trained LLMs for time-series forecasting. Moirai (Woo et al., 2024) introduces a novel transformer architecture and a related large dataset of time series from several domains to support the requirements of universal forecasting.

## 3 MODEL ARCHITECTURE

The forecasting problem for multivariate time-series can be encoded as follows: given an input multivariate time series $\boldsymbol{X} = \{\boldsymbol{x}_1, \cdots, \boldsymbol{x}_L\} \in \mathbb{R}^{L \times C}$ where $L$ is the input lookback window and $C$ is the number of channels, we want to predict the future $T$ time steps $\{\boldsymbol{x}_{L+1}, \cdots, \boldsymbol{x}_{L+T}\} \in \mathbb{R}^{T \times C}$.

Figure 1 depicts our model architecture. We employ a fully transformer-based architecture, incorporating a multi-patch attention mechanism.

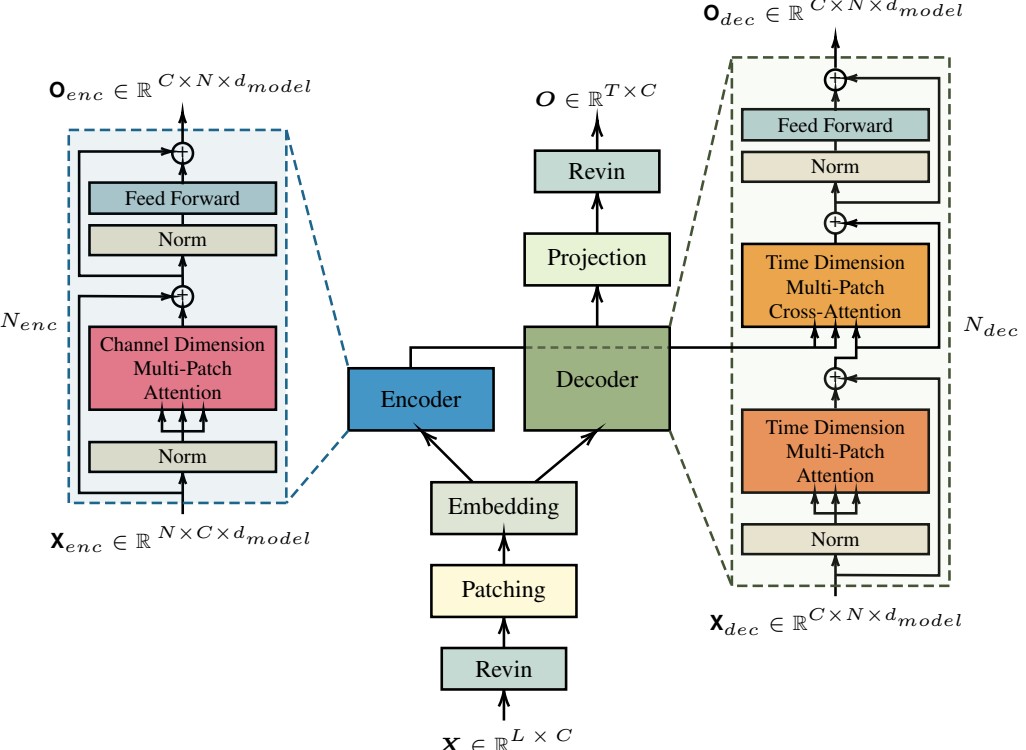

Figure 1: Sentinel architecture

To mitigate the distributional shift problem, we integrate RevIN (Reversible Instance Normalization) (Kim et al., 2022). RevIN normalizes the input sequence using instance-specific mean and standard deviation. During normalization, learnable affine parameters ($\gamma$ and $\beta$) are applied to scale and shift the input, allowing the model to process inputs with consistent statistics. At the output stage,

a corresponding denormalization step reverses the process, restoring predictions to their original scale.

## 3.1 Patching and Embedding

Following the normalization process described earlier, the input time series undergoes a patching process (Nie et al., 2023) which divides the input series into patches of size $P$ along the time dimension. This patching is performed with overlapping regions determined by the stride $S$. After this operation, the resulting tensor is represented as $\overline{\mathbf{X}} \in \mathbb{R}^{C \times N \times P}$ where $N = [\frac{L-P}{S} + 1]$ is the number of patches.

After patching, the patched tensor is projected to the model's latent dimension resulting in the embedding tensor $\mathbf{X}^{(1)} \in \mathbb{R}^{C \times N \times d_{model}}$. This transformation is performed through a multi-layer perceptron (MLP).

As highlighted by Nie et al. (2023), since the computational complexity of the attention mechanism scales quadratically with the length of the sequence, dividing the input time series into smaller patches, the memory usage and computational complexity are reduced quadratically by a factor $S$.

## 3.2 Reshaping

The tensors for encoder and decoder are shaped differently from each other, allowing the model to focus on distinct aspects of the time series data. Given the tensor $\mathbf{X}^{(1)} \in \mathbb{R}^{C \times N \times d_{model}}$ we perform the following shaping

$$\mathbf{X}_{enc} = \mathbf{X}^{(2)} \in \mathbb{R}^{N \times C \times d_{model}} \tag{1}$$

$$\mathbf{X}_{dec} = \mathbf{X}^{(1)} \in \mathbb{R}^{C \times N \times d_{model}} \tag{2}$$

This reshaping enables the encoder to focus on inter-channel relationships and the decoder to focus on temporal dependencies, thus enhancing the model's ability to capture both types of relationships effectively.

## 3.3 Multi-Patch Attention

The patching process introduces a natural link to the multi-head attention mechanism, where the multi-head splitting can be reinterpreted as a multi-patch splitting. This enables us to modify the attention structure leveraging the patching operation to efficiently capture diverse patterns in the input sequence.

In the traditional multi-head transformer architecture (Vaswani et al., 2017), the Query ($\boldsymbol{Q}$), Key ($\boldsymbol{K}$) and Value ($\mathbf{V}$) are projected into $h$ independent attention heads. Each attention head $i$ has its own set of learned projection matrices $\boldsymbol{W}_i^Q, \boldsymbol{W}_i^K, \boldsymbol{W}_i^V \in \mathbb{R}^{d_{model} \times d_h}$, where $d_h = \frac{d_{model}}{h}$ is the dimension of each head.

Given an input matrix $\boldsymbol{X} \in \mathbb{R}^{L \times d_{model}}$, the corresponding query, key, and value representations for each head are computed as follows:

$$\boldsymbol{Q}_i = \boldsymbol{X}\boldsymbol{W}_i^Q, \quad \boldsymbol{K}_i = \boldsymbol{X}\boldsymbol{W}_i^K, \quad \boldsymbol{V}_i = \boldsymbol{X}\boldsymbol{W}_i^V \tag{3}$$

$\boldsymbol{Q}_i, \boldsymbol{K}_i, \boldsymbol{V}_i \in \mathbb{R}^{L \times d_h}$ are the projected query, key and value matrices for the $i$-th attention head. Each head independently refers to different sub-spaces of the input sequence, enabling multi-head attention to capture different relations along the input sequence. The attention output is computed as:

$$MultiHead(\boldsymbol{Q}, \boldsymbol{K}, \boldsymbol{V}) = Concat(head_1, \cdots, head_h)\boldsymbol{W}^O \in \mathbb{R}^{L \times d_{model}} \tag{4}$$

$$head_i = Attention(\boldsymbol{X}\boldsymbol{W}_i^Q, \boldsymbol{X}\boldsymbol{W}_i^K, \boldsymbol{X}\boldsymbol{W}_i^V) \in \mathbb{R}^{L \times d_h} \tag{5}$$

$$Attention(\boldsymbol{Q}_i, \boldsymbol{K}_i, \boldsymbol{V}_i) = \text{softmax}\left(\frac{\boldsymbol{Q}_i \boldsymbol{K}_i^T}{\sqrt{d_h}}\right) \boldsymbol{V}_i \in \mathbb{R}^{L \times d_h} \tag{6}$$

where $\boldsymbol{W}^O \in \mathbb{R}^{hd_h \times d_{model}}$ is the learned weight matrix that projects the concatenated outputs back to $\mathbb{R}^{L \times d_{model}}$.

In contrast, in our proposed Multi-Patch Attention, we eliminate the multi-head splitting by leveraging the patch-based input structure. Instead of splitting the input into multiple heads, attention is applied directly to each patch. Furthermore, since the encoder deals with channel attention and the decoder with temporal attention, we use a reshaping function (see Section 3.2) to prepare the two inputs.

Considering the encoder input $\mathbf{X}^{(2)} \in \mathbb{R}^{N \times C \times d_{model}}$, we can extract $N$ patches $\boldsymbol{X}_n^{(2)} \in \mathbb{R}^{C \times d_{model}}$, $n = 1, \cdots, N$ by slicing on the first dimension. For each patch we can then compute matrices $\boldsymbol{Q}_n^{(2)}, \boldsymbol{K}_n^{(2)}, \boldsymbol{V}_n^{(2)}$ using the following equation:

$$\boldsymbol{Q}_n^{(2)} = \boldsymbol{X}_n^{(2)} \boldsymbol{W}^Q, \quad \boldsymbol{K}_n^{(2)} = \boldsymbol{X}_n^{(2)} \boldsymbol{W}^K, \quad \boldsymbol{V}_n^{(2)} = \boldsymbol{X}_n^{(2)} \boldsymbol{W}^V \tag{7}$$

where, $\boldsymbol{W}^Q, \boldsymbol{W}^K, \boldsymbol{W}^V \in \mathbb{R}^{d_{model} \times d_{model}}$ are learned weight matrices shared across all patches and $\boldsymbol{Q}_n^{(2)}, \boldsymbol{K}_n^{(2)}, \boldsymbol{V}_n^{(2)} \in \mathbb{R}^{C \times d_{model}}$.

This is different than the classical multi-head mechanism, in which a single input is multiplied by specific projection matrices $\boldsymbol{W}_i^Q, \boldsymbol{W}_i^K, \boldsymbol{W}_i^V$ (see equation 3). In the multi-patch attention, instead, different input patches $\boldsymbol{X}_n^{(2)}$, $n = 1, \cdots, N$ are multiplied by the same projection matrices $\boldsymbol{W}^Q, \boldsymbol{W}^W, \boldsymbol{W}^V$ (see equation 7). Figure 2 shows how the patching process can be exploited to create a structure similar to the one created through the multi-head splitting. The multi-patch attention is computed in the encoder according to the following formulas:

$$MultiPatch(\boldsymbol{Q}^{(2)}, \boldsymbol{K}^{(2)}, \boldsymbol{V}^{(2)}) = Fold(patch_1, \cdots, patch_N) \in \mathbb{R}^{N \times C \times d_{model}} \tag{8}$$

$$patch_n = Attention(\boldsymbol{X}_n^{(2)} \boldsymbol{W}^Q, \boldsymbol{X}_n^{(2)} \boldsymbol{W}^K, \boldsymbol{X}_n^{(2)} \boldsymbol{W}^V) \in \mathbb{R}^{C \times d_{model}} \tag{9}$$

$$Attention(\boldsymbol{Q}_n^{(2)}, \boldsymbol{K}_n^{(2)}, \boldsymbol{V}_n^{(2)}) = \text{softmax}\left(\frac{\boldsymbol{Q}_n^{(2)} \boldsymbol{K}_n^{(2)T}}{\sqrt{d_{model}}}\right) \boldsymbol{V}_n \in \mathbb{R}^{C \times d_{model}} \tag{10}$$

In the decoder, the equations remain the same but the input tensor is $\mathbf{X}^{(1)} \in \mathbb{R}^{C \times N \times d_{model}}$ instead of $\mathbf{X}^{(2)} \in \mathbb{R}^{N \times C \times d_{model}}$, hence we extract $C$ patches $patch_c$, $c = 1, \cdots, C$ instead of $N$ patches, and query, key and value matrices are $\boldsymbol{Q}_c^{(1)}, \boldsymbol{K}_c^{(1)}, \boldsymbol{V}_c^{(1)} \in \mathbb{R}^{N \times d_{model}}$. Consequently, $Attention(\boldsymbol{Q}_c^{(1)}, \boldsymbol{K}_c^{(1)}, \boldsymbol{V}_c^{(1)}) \in \mathbb{R}^{N \times d_{model}}$ and $MultiPatch(\boldsymbol{Q}^{(1)}, \boldsymbol{K}^{(1)}, \boldsymbol{V}^{(1)}) \in \mathbb{R}^{C \times N \times d_{model}}$.

### 3.4 Encoder and Decoder

**Encoder** Attention across channels was shown to be effective in various works (Liu et al., 2024; Wang et al., 2024; Zhang & Yan, 2023). In our model, the encoder receives as input

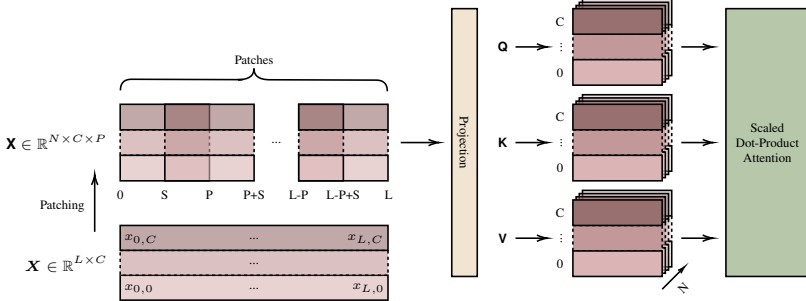

Figure 2: The figure illustrates the multi-patch attention mechanism. Initially, the time series is divided into multiple patches, where $N$ represents the number of patches and $P$ denotes the patch size. Each patch is generated with a stride $S$, which defines the distance between consecutive patches. On the right-hand side, the figure shows how this patching structure can be seamlessly integrated into a multi-head attention mechanism. By leveraging the patch-based representation, the patches serve as inputs to the attention layer, effectively exploiting the structure naturally induced by the patching operation.

$\mathbf{X}^{(2)} \in \mathbb{R}^{N \times C \times d_{model}}$, enabling the model to learn relationships between the $C$ different channels across the sequence. In the Scaled Dot-Product Attention we compute the attention between channels, namely,

$$\boldsymbol{W}_{attn}^{(2)} = \mathrm{softmax}\left(\frac{\boldsymbol{Q}^{(2)}\boldsymbol{K}^{(2)T}}{\sqrt{d_k}}\right)$$

where $\mathbf{Q}^{(2)}, \mathbf{K}^{(2)} \in \mathbb{R}^{N \times C \times d_{model}}$. The resulting attention weights $\mathbf{W}_{attn}^{(2)} \in \mathbb{R}^{N \times C \times C}$, show how each channel relates to every other channel in each patch $n = 1, \ldots, N$, enabling the attention mechanism to focus on the relationships between channels within a patch. Finally, the attention weights are applied to the value tensor, i.e., $\boldsymbol{W}_{attn}^{(2)}\mathbf{V}^{(2)}$ resulting in the attention output $\in \mathbb{R}^{N \times C \times d_{model}}$.

The output of the encoder is then reshaped from $\mathbb{R}^{N \times C \times d_{model}}$ to $\mathbb{R}^{C \times N \times d_{model}}$ resulting in $\mathbf{O}_{enc} \in \mathbb{R}^{C \times N \times d_{model}}$ (see Figure 1) and used as the key (K) and value (V) in the cross-attention part of the decoder.

**Decoder**   The decoder receives the input tensor $\mathbf{X}^{(1)} \in \mathbb{R}^{C \times N \times d_{model}}$. Both self-attention and cross-attention in the decoder are applied along the temporal dimension to capture causal temporal dependencies

In the self-attention mechanism, the query $\mathbf{Q}^{(1)}$, key $\mathbf{K}^{(1)}$ and value $\mathbf{V}^{(1)}$ tensors are derived from $\mathbf{X}^{(1)}$ and the attention is performed along the temporal dimension with causal masking applied. This allows the model to learn relationships between different time steps, capturing temporal patterns across the sequence.

In the cross-attention mechanism, the query $\mathbf{Q}$ is the output of the decoder self-attention operation, while the key $\mathbf{K}^{O_{enc}}$ and value $\mathbf{V}^{O_{enc}}$ are derived from the encoder output $\mathbf{O}_{enc}$ which encapsulates relationships across channels.

By performing cross-attention over the temporal dimension, the decoder integrates both temporal dependencies (from the self-attention mechanism) and cross-channel information (from the encoder), effectively merging both channel and temporal relationships in the final output.

## 4   EXPERIMENTS

In this section, we introduce the datasets used to evaluate our approach, along with the baseline methods selected for comparison. Before presenting the results, we will discuss the experimental settings adopted to achieve these outcomes. Following this, we will present the results, including

an ablation study that highlights the importance of the proposed components. Additionally, we will demonstrate how performance improves as the lookback window increases.

## 4.1 EXPERIMENTAL SETTING

**Datasets**    We evaluated the performance of our model on various benchmarks used for long-term forecasting, including ETTh1, ETTh2, ETTm1, ETTm2, Electricity, and Weather. Details about the number of variables of each dataset, their training/validation/test sizes and frequency are available in Table 1.

| Datasets | Variables | Dataset Size | Frequency |
|----------|-----------|--------------|-----------|
| ETTh1 | 7 | (8545, 2881, 2881) | Hourly |
| ETTh2 | 7 | (8545, 2881, 2881) | Hourly |
| ETTm1 | 7 | (34465, 11521, 11521) | 15 Minutes |
| ETTm2 | 7 | (34465, 11521, 11521) | 15 Minutes |
| Weather | 21 | (36792, 5271, 10540) | 10 Minutes |
| Electricity | 321 | (18317, 2633, 5261) | Hourly |
| Traffic | 862 | (12185, 1757, 3509) | Hourly |

Table 1: Description of the datasets

**Baselines**    To evaluate the performance of Sentinel, we selected the following forecasting models as benchmarks, based on their good performance. Specifically, we used: CARD (Wang et al., 2024), iTransformer (Liu et al., 2024), PatchTST(Nie et al., 2023), Crossformer (Zhang & Yan, 2023), DLinear (Zeng et al., 2023), FEDformer (Zhou et al., 2022), Autoformer (Wu et al., 2021).

**Settings**    We run the experiments on each dataset with multiple seeds and report the average Mean Squared Error (MSE) and Mean Absolute Error (MAE). In all experiments, we used a patch size of 16 and a stride of 8. The dropout is set to 0.3 and AdamW (Loshchilov & Hutter, 2019) is used as optimizer with a learning rate of 0.0005 and an L1 loss. For each dataset we perform various runs with a variable number of encoder layers $N_{enc} = 1, \cdots, 4$ (see Figure 1), decoder layers $N_{dec} = 1, \cdots, 4$ (see Figure 1), and different model dimension $d_{model} = 16, 32, 64, 128, 256, 512$. We selected the configuration yielding the best performance. To align with baseline performance in the literature, we used a fixed lookback window $L = 96$ and a set of different prediction length $T = (96, 192, 336, 720)$. The model parameters are consistent across all prediction lengths within each dataset.

**Hardware and software**    We run our experiments on a NVIDIA RTX A6000 (48 GiB) and on a on a NVIDIA A100 (80 GiB). The code is attached to this submission and it will be published online upon acceptance.

## 4.2 RESULTS

The results, summarized in Table 2, show that Sentinel is consistently the best or second-best performing approach among all competitors. Alongside CARD (Wang et al., 2024), Sentinel demonstrates superior forecasting accuracy compared to the other models. In general, we observe that models incorporating specialized blocks to handle the channel dimension, such as iTransformer (Liu et al., 2024) and CARD (Wang et al., 2024), tend to outperform those without such component. However, since the temporal dimension is equally critical, approaches like Sentinel and CARD, which integrates components specifically designed to manage both channel anpatch tsd temporal dependecies, consistently deliver the hightest performance. This highlights the importance of effectively addressing both channel and temporal dimensions in multivariate time series forecasting, a balance that Sentinel achieves through its specialized architecture.

| Models | | Sentinel | | CARD | | iTransformer | | PatchTST | | Crossformer | | DLinear | | FEDformer | | Autoformer | |
|---|---|---|---|---|---|---|---|---|---|---|---|---|---|---|---|---|---|
| Metric | | MSE | MAE | MSE | MAE | MSE | MAE | MSE | MAE | MSE | MAE | MSE | MAE | MSE | MAE | MSE | MAE |
| ETTm1 | 96 | **0.311** | **0.340** | 0.316 | 0.347 | 0.334 | 0.368 | 0.342 | 0.378 | 0.366 | 0.400 | 0.345 | 0.372 | 0.764 | 0.416 | 0.505 | 0.475 |
| | 192 | **0.360** | **0.368** | 0.363 | 0.370 | 0.377 | 0.391 | 0.372 | 0.393 | 0.396 | 0.414 | 0.380 | 0.389 | 0.426 | 0.441 | 0.553 | 0.496 |
| | 336 | **0.391** | 0.392 | 0.392 | **0.390** | 0.426 | 0.420 | 0.402 | 0.413 | 0.439 | 0.443 | 0.413 | 0.413 | 0.445 | 0.459 | 0.621 | 0.537 |
| | 720 | **0.456** | 0.432 | 0.458 | **0.425** | 0.491 | 0.459 | 0.642 | 0.449 | 0.540 | 0.509 | 0.474 | 0.453 | 0.543 | 0.490 | 0.671 | 0.561 |
| | Avg | **0.379** | **0.383** | 0.383 | 0.384 | 0.407 | 0.410 | 0.395 | 0.408 | 0.435 | 0.417 | 0.403 | 0.407 | 0.448 | 0.452 | 0.588 | 0.517 |
| ETTm2 | 96 | 0.172 | **0.246** | **0.169** | 0.248 | 0.180 | 0.264 | 0.176 | 0.258 | 0.273 | 0.346 | 0.193 | 0.292 | 0.203 | 0.287 | 0.255 | 0.339 |
| | 192 | 0.238 | 0.295 | **0.234** | **0.292** | 0.250 | 0.309 | 0.244 | 0.304 | 0.350 | 0.421 | 0.284 | 0.362 | 0.296 | 0.328 | 0.281 | 0.340 |
| | 336 | 0.301 | **0.335** | **0.294** | 0.339 | 0.311 | 0.348 | 0.304 | 0.342 | 0.474 | 0.505 | 0.369 | 0.427 | 0.325 | 0.366 | 0.339 | 0.372 |
| | 720 | 0.401 | 0.391 | **0.390** | **0.388** | 0.412 | 0.407 | 0.408 | 0.403 | 1.347 | 0.812 | 0.554 | 0.522 | 0.421 | 0.415 | 0.433 | 0.432 |
| | Avg | 0.278 | **0.317** | **0.272** | 0.317 | 0.288 | 0.332 | 0.283 | 0.327 | 0.609 | 0.521 | 0.350 | 0.401 | 0.305 | 0.349 | 0.327 | 0.371 |
| ETTh1 | 96 | 0.390 | 0.396 | 0.383 | **0.391** | 0.386 | 0.405 | 0.426 | 0.426 | 0.391 | 0.417 | 0.386 | 0.400 | **0.376** | 0.419 | 0.449 | 0.459 |
| | 192 | 0.445 | 0.425 | 0.435 | **0.420** | 0.441 | 0.436 | 0.469 | 0.452 | 0.449 | 0.452 | 0.437 | 0.432 | **0.420** | 0.448 | 0.500 | 0.482 |
| | 336 | 0.490 | 0.447 | 0.479 | **0.442** | 0.487 | 0.458 | 0.506 | 0.473 | 0.510 | 0.489 | 0.481 | 0.459 | **0.459** | 0.465 | 0.521 | 0.496 |
| | 720 | 0.490 | 0.468 | **0.471** | **0.461** | 0.503 | 0.491 | 0.504 | 0.495 | 0.594 | 0.567 | 0.519 | 0.516 | 0.506 | 0.507 | 0.514 | 0.512 |
| | Avg | 0.453 | 0.434 | 0.442 | **0.429** | 0.454 | 0.447 | 0.455 | 0.444 | 0.486 | 0.481 | 0.456 | 0.452 | **0.440** | 0.460 | 0.496 | 0.487 |
| ETTh2 | 96 | 0.299 | **0.320** | **0.281** | 0.330 | 0.297 | 0.349 | 0.292 | 0.342 | 0.641 | 0.549 | 0.333 | 0.387 | 0.358 | 0.397 | 0.346 | 0.388 |
| | 192 | 0.370 | 0.386 | **0.363** | **0.381** | 0.380 | 0.400 | 0.387 | 0.400 | 0.896 | 0.656 | 0.477 | 0.476 | 0.429 | 0.439 | 0.456 | 0.452 |
| | 336 | 0.417 | **0.418** | **0.411** | 0.418 | 0.428 | 0.432 | 0.426 | 0.434 | 0.936 | 0.690 | 0.594 | 0.541 | 0.496 | 0.487 | 0.482 | 0.486 |
| | 720 | 0.425 | 0.439 | **0.416** | **0.431** | 0.427 | 0.445 | 0.430 | 0.446 | 1.390 | 0.863 | 0.831 | 0.657 | 0.463 | 0.474 | 0.515 | 0.511 |
| | Avg | 0.378 | 0.391 | **0.368** | **0.390** | 0.383 | 0.407 | 0.384 | 0.406 | 0.966 | 0.690 | 0.559 | 0.515 | 0.437 | 0.449 | 0.450 | 0.459 |
| Weather | 96 | 0.161 | 0.195 | **0.150** | **0.188** | 0.174 | 0.214 | 0.176 | 0.218 | 0.164 | 0.232 | 0.196 | 0.255 | 0.217 | 0.296 | 0.266 | 0.336 |
| | 192 | 0.210 | 0.243 | **0.202** | **0.238** | 0.221 | 0.254 | 0.223 | 0.259 | 0.211 | 0.276 | 0.237 | 0.296 | 0.276 | 0.336 | 0.307 | 0.367 |
| | 336 | 0.266 | 0.284 | **0.260** | **0.282** | 0.278 | 0.296 | 0.277 | 0.297 | 0.269 | 0.327 | 0.283 | 0.335 | 0.339 | 0.380 | 0.359 | 0.395 |
| | 720 | **0.342** | **0.337** | 0.343 | 0.353 | 0.358 | 0.347 | 0.353 | 0.347 | 0.355 | 0.404 | 0.345 | 0.381 | 0.403 | 0.428 | 0.419 | 0.428 |
| | Avg | 0.244 | 0.265 | **0.239** | **0.261** | 0.258 | 0.278 | 0.257 | 0.280 | 0.250 | 0.310 | 0.265 | 0.317 | 0.309 | 0.360 | 0.338 | 0.382 |
| Electricity | 96 | **0.140** | 0.235 | 0.141 | **0.233** | 0.148 | 0.240 | 0.190 | 0.296 | 0.254 | 0.347 | 0.197 | 0.282 | 0.193 | 0.308 | 0.201 | 0.317 |
| | 192 | 0.162 | **0.250** | **0.160** | 0.250 | 0.162 | 0.253 | 0.199 | 0.304 | 0.261 | 0.353 | 0.196 | 0.285 | 0.201 | 0.345 | 0.222 | 0.334 |
| | 336 | 0.181 | 0.268 | **0.173** | **0.263** | 0.178 | 0.269 | 0.217 | 0.319 | 0.273 | 0.364 | 0.209 | 0.301 | 0.214 | 0.329 | 0.231 | 0.338 |
| | 720 | 0.219 | 0.303 | **0.197** | **0.284** | 0.225 | 0.317 | 0.258 | 0.352 | 0.303 | 0.388 | 0.245 | 0.333 | 0.246 | 0.355 | 0.254 | 0.361 |
| | Avg | 0.176 | 0.264 | **0.168** | **0.258** | 0.178 | 0.270 | 0.216 | 0.318 | 0.273 | 0.363 | 0.212 | 0.300 | 0.214 | 0.327 | 0.227 | 0.338 |
| Traffic | 96 | 0.478 | **0.263** | 0.419 | 0.269 | **0.395** | 0.268 | 0.462 | 0.315 | 0.558 | 0.320 | 0.650 | 0.396 | 0.587 | 0.366 | 0.613 | 0.388 |
| | 192 | 0.481 | **0.270** | 0.443 | 0.276 | **0.417** | 0.276 | 0.473 | 0.321 | 0.569 | 0.321 | 0.650 | 0.396 | 0.604 | 0.373 | 0.616 | 0.382 |
| | 336 | 0.496 | **0.253** | 0.460 | 0.283 | **0.433** | 0.283 | 0.494 | 0.331 | 0.591 | 0.328 | 0.605 | 0.373 | 0.621 | 0.383 | 0.622 | 0.337 |
| | 720 | 0.547 | **0.294** | 0.490 | 0.299 | **0.467** | 0.302 | 0.522 | 0.342 | 0.652 | 0.359 | 0.650 | 0.396 | 0.626 | 0.382 | 0.660 | 0.408 |
| | Avg | 0.501 | **0.270** | 0.453 | 0.282 | **0.428** | 0.282 | 0.488 | 0.327 | 0.593 | 0.332 | 0.625 | 0.383 | 0.610 | 0.376 | 0.628 | 0.379 |

Table 2: Results table for long-term forecasting tasks. The lookback window is set as 96 for all experiments with a varying prediction horizons 96, 192, 336, 720. Avg is the average of all four predictions lengths. Each dataset is run on multiple seeds. The best models is indicated in bold, the second underlined. All reported results are extracted from CARD (Wang et al., 2024), except for the iTransformer (Liu et al., 2024), which were obtained directly from its original paper.

## 4.3 ABLATION-STUDY

We conducted an ablation study by systematically removing the different components proposed in this paper, with the results summarized in Table 3. As seen in the table, each component plays a critical role in achieving high performance. Specifically, the benefits introduced by Sentinel are more pronounced in datasets with a large number of features. For instance, in datasets with many channels, such as electricity or weather in Table 3, the removal of the channel-encoder, responsible

for handling cross-channel relationships, leads to a significant drop in performance (i.e., respectively $-8.0\%$ and $-9.7\%$ of MSE; $-3.0\%$ and $-4.5\%$ in terms of MSE). Conversely, in datasets with fewer features, such as ETTh2 in Table 3, the impact is less severe. In these scenarios, the most harmful effect is observed when the decoder is removed (i.e., $-0.3\%$ of MSE and $-1.3\%$ of MAE), as the reduced feature set places less emphasis on feature importance, making the role of the decoder more critical.

Lastly, the final row of the table, shows the architecture using the classic multi-head encoder and decoder, without the multi-patch attention. It consistently performs lower than the multi-patch attention solution across all datasets, regardless of their properties. This strong and consistent performance gap highlights the importance of rethinking the traditional "head" attention into a "patch" attention. By replacing multi-head attention with our novel multi-patch attention, we exploit the natural structure of the time-series created by the patching process, thereby increasing the overall forecasting performance. All these results confirm the importance of the Sentinel components for effectively processing multivariate datasets and providing improved forecasting performance.

| Components | | Electricity | | | | Weather | | | | ETTh2 | | | |
|---|---|---|---|---|---|---|---|---|---|---|---|---|---|
| Encoder | Decoder | MSE | % | MAE | % | MSE | % | MAE | % | MSE | % | MAE | % |
| Multi-Patch Channel | Multi-Patch Time | **0,176** | - | **0,263** | - | **0,244** | - | **0,264** | - | 0,378 | - | **0,391** | - |
| Multi-Patch Time | Multi-Patch Time | 0,190 | -8,0 | 0,271 | -3,0 | 0,249 | -2,1 | 0,267 | -1,1 | 0,379 | -0,3 | 0,395 | -1,0 |
| Multi-Patch Channel | w/o Decoder | 0,182 | -3,4 | 0,269 | -2,2 | 0,249 | -2,1 | 0,270 | -2,3 | 0,379 | -0,3 | 0,396 | -1,3 |
| w/o Encoder | Multi-Patch Time | 0,193 | -9,7 | 0,275 | -4,5 | 0,250 | -2,5 | 0,269 | -1,9 | **0,375** | 0,8 | 0,393 | -0,5 |
| Multi-Head Channel | Multi-Head Time | 0,180 | -2,3 | 0,265 | -0,8 | 0,249 | -2,1 | 0,267 | -1,1 | 0,382 | -1,1 | 0,399 | -2,1 |

Table 3: Ablation study results for various dataset configurations. We conducted experiments on various datasets, replacing or removing components proposed in this work. The first row indicates the configuration of Sentinel. The percentage column reflects the relative performance degradation for each configuration compared to the proposed approach.

## 4.4 VARYING-LOOKBACK WINDOW

Transformer-based architectures have been shown to experience performance degradation as the lookback window increases (Zeng et al., 2023). In our evaluation, as in other works (Wang et al., 2024; Liu et al., 2024; Nie et al., 2023) we explored the impact of varying lookback lengths. As shown in Figure 3 and Tables 4.4, our approach demonstrates that increasing lookback window, thereby providing the model with more historical information for forecasting, generally leads to improved overall performance. This suggests that a longer historical context can enhance the model's ability to capture temporal patterns and make more accurate predictions.

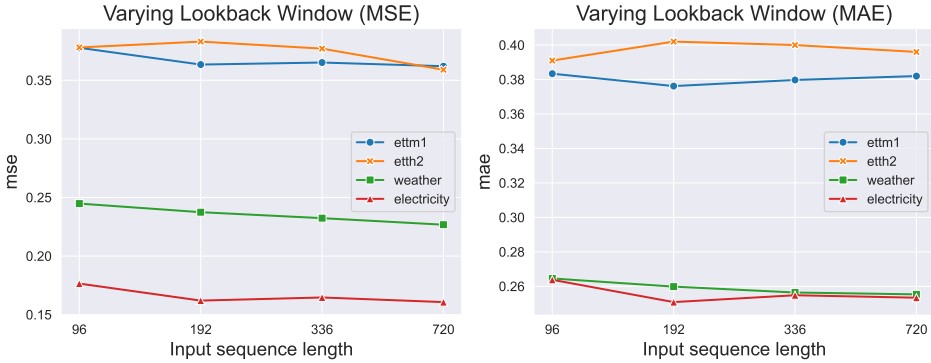

Figure 3: Evolution of MAE and MSE Based on Lookback Window

| Dataset | MSE | | | | Dataset | MAE | | | |
|---|---|---|---|---|---|---|---|---|---|
| | Lookback Window | | | | | Lookback Window | | | |
| | 96 | 196 | 336 | 720 | | 96 | 196 | 336 | 720 |
| ETTh2 | 0,378 | 0,383 | 0,377 | **0,359** | ETTh2 | **0,391** | 0,402 | 0,400 | 0,396 |
| ETTm1 | 0,377 | 0,363 | 0,365 | **0,362** | ETTm1 | 0,383 | **0,376** | 0,379 | 0,382 |
| Weather | 0,244 | 0,237 | 0,232 | **0,226** | Weather | 0,264 | 0,259 | 0,256 | **0,255** |
| Electricity | 0,176 | 0,161 | 0,164 | **0,160** | Electricity | 0,263 | **0,250** | 0,254 | 0,253 |

Table 4: The two tables show the variation of MSE and MAE with respect to the lookback window. Values in the tables represent the means calculated over multiple prediction lengths $(96, 192, 336, 720)$

## 5 CONCLUSION AND FUTURE WORK

In this paper, we introduced Sentinel, a fully Transformer-based architecture specifically designed for multivariate time-series forecasting. Sentinel addresses a critical limitation in existing transformer models by simultaneously capturing both cross-channel and temporal dependencies, which are essential for improving forecast accuracy in complex multivariate datasets. Additionally, we introduced the concept of multi-patch attention to leverage the structure created through the patching process. This novel component has proven to be highly valuable in ablation studies, significantly enhancing forecasting performance. By rethinking the traditional multi-head attention and replacing it with multi-patch attention, Sentinel is able to better utilize the patched time-series structure, leading to superior results across a wide range of multivariate datasets.

In the near future, we aim to refine the architecture of Sentinel by focusing on both the encoder and the decoder. As highlighted in the ablation study, the encoder shows a strong impact when dealing with datasets with a high number of features but it tends to overfit when the number of features is lower. To address this lack, we aim to develop techniques to mitigate overfitting in such cases, ensuring the encoder remains effective across various sizes of the feature-space. Conversely, the decoder has less impact as the number of features increases. We aim to improve its contribution in such high-feature scenarios. Another promising direction is to evaluate the few-shot learning capabilities of Sentinel, which could expand its applicability to a wider range of forecasting tasks.

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
