# OpenReview forum: "Sentinel: Multi-Patch Transformer with Temporal and Channel Attention for Time Series Forecasting"
_ICLR.cc/2025/Conference — Submitted to ICLR 2025_

### Official Review · Reviewer_Qk9P · 2024-10-31

**Soundness:** 2
**Presentation:** 2
**Contribution:** 1
**Rating:** 3
**Confidence:** 4

**Summary:**

The paper proposed a Transformer-based model designed for multivariate time-series forecasting. The model introduces a multi-patch attention mechanism that provides two sets of patches, one is from temporal direction and other is from inter-channel direction. Subsequently, the use decoder and decoder to model them separately.
This paper is very clear and easy to understand.

**Strengths:**

1. A useful insight is that the authors link multi-head splitting to multi-patch splitting, seamlessly introducing their new attention mechanism.
2. I agree that the ablation study is reasonable.
3. The paper includes comparisons with numerous baselines.

**Weaknesses:**

1. The main concern is that the proposed method does not demonstrate sufficient power, as evident from the experimental results. CARD clearly outperforms the proposed method on numerous datasets. This undermines the authors’ claim, as Sentinel introduces more complexity than CARD with added components—such as “specializing the encoder in capturing contextual information through the channel dimension.” As a result, the contribution may not be sufficient, as integrating inter-channel information effectively could likely boost independent Transformers (PatchTST).

2. Although the authors review many recent papers, they fail to clearly explain their model’s advantage over existing approaches.


3. Compared to similar ICLR papers, the experiments seem insufficient. For instance, CARD, iTransformer, and PatchTST conducted more extensive experiments, either across more datasets or with more comprehensive experimental settings.

**Questions:**

What are the advantages of the proposed method compared to CARD and other methods? For example, does it offer better efficiency or improved performance with limited data?

---

> ### Author Response · Authors · 2024-11-21
>
> - The main concern is that the proposed method does not demonstrate sufficient power: We respectfully disagree with the assertion that Sentinel adds more complexity than CARD. Sentinel employs a specific attention structure that is seamlessly aligned with the patching process during patch splitting. Its design leverages the original Transformer architecture, with the encoder using a non-causal mask to freely capture inter-channel dependencies across the sequence and the decoder employing a causal mask to model temporal relationships, ensuring that future values do not influence past ones. This is achieved through simple tensor shape manipulations without altering the core Transformer architecture. By contrast, CARD introduces a modified attention block and a reshaping process with additional elements such as a token blend module and a custom loss function. Regarding the suggestion to adapt PatchTST for channel dependency, it is important to note that PatchTST was intentionally designed with channel independence as its core principle. Transitioning PatchTST to incorporate channel dependency would contradict its fundamental design philosophy and introduce significant technical challenges. Furthermore, the feasibility of achieving improved results with such an adaptation remains unproven and would require extensive validation.

---

> > ### Comment · Reviewer_Qk9P · 2024-11-21
> >
> > Thanks for the response. I partially agree with the authors' points. CARD introduces an additional loss and token to enhance its modeling capabilities, which may be complicate in implementation. However, it uses an encoder-only structure to capture both temporal and cross-channel characteristics, aligning with the mainstream methods that also favor encoder-only architectures. From this perspective, I believe the encoder-only structure has its own merits. I suggest the authors consider discussing this aspect further if convenient.
> >
> > That said, after reviewing the feedback from other reviewers, I still find the novelty of the work potentially insufficient for acceptance at ICLR. For now, I will maintain my score but may adjust it based on further input from other reviewers.

---

### Official Review · Reviewer_VQH7 · 2024-11-01

**Soundness:** 2
**Presentation:** 2
**Contribution:** 2
**Rating:** 3
**Confidence:** 4

**Summary:**

The paper proposes an encoder-decoder architecture for time series forecasting that models the channel dimension and temporal dimension separately. It introduces a multi-patch attention mechanism to replace the standard multi-head attention to enhance the prediction performance.​

**Strengths:**

- This paper considers both channel and temporal dimensions in the modeling of time series.

- This paper proposes a multi-patch attention mechanism to replace the standard Transformer multi-head attention by focusing on patch-level information in time series.

**Weaknesses:**

The motivation of the method lacks clarity, and its novelty is limited. Furthermore, the performance of the proposed method does not show a clear advantage over other baselines, and there is a lack of validation of the rationale behind the design of individual model components.

**Questions:**

- About the method: The approach of modeling the channel and time dimensions separately lacks novelty. The proposed multi-patch attention mechanism does not convincingly demonstrate effectiveness compared to the standard Transformer structure, nor does it show significant performance gains.

- About main experiments: The primary experiments do not show a clear advantage over existing methods, lacking of recent baselines such as TimeXer and TimeMixer.

- About analytical experiments: Since the authors claim that a key innovation of the paper is the multi-patch attention mechanism’s application to the channel and time dimensions, detailed analytical experiments are necessary. For instance, visualizations showing whether the model has learned meaningful attention patterns could substantiate the rationale and effectiveness of the design.

[1] TimeXer: Empowering Transformers for Time Series Forecasting with Exogenous Variables

[2] TimeMixer: Decomposable Multiscale Mixing for Time Series Forecasting

---

> ### Author Response · Authors · 2024-11-21
>
> - TimeXer and TimeMixer: Thank you for your suggestion regarding additional baselines. We can include TimeMixer in our experiments to further strengthen our comparisons. However, TimeXer is currently an unpublished article, and its code is unavailable.

---

> > ### Comment · Reviewer_VQH7 · 2024-11-22
> >
> > You can find the open-sourced code of TimeXer in https://arxiv.org/pdf/2402.19072

---

> ### Comment · Reviewer_VQH7 · 2024-11-26
>
> Due to the limited novelty and absence of significant performance advantages, I will maintain my score.

---

### Official Review · Reviewer_NEuV · 2024-11-03

**Soundness:** 2
**Presentation:** 2
**Contribution:** 2
**Rating:** 3
**Confidence:** 4

**Summary:**

The author proposed an encoder-decoder model structure that can simultaneously capture temporal relationships and variable relationships. Additionally, the author employed methods such as multi-patch to enhance the model's performance, achieving comparable performance on benchmark datasets by addressing both dependency types effectively.

**Strengths:**

1. its ability to capture both temporal and inter-channel dependencies crucial for multivariate time series forecasting.
2. by utilizing a patching process, Sentinel introduces a new attention mechanism that replaces traditional multi-head attention splitting. This shifts the focus from "heads" to "patches," integrating more naturally into the Transformer architecture.
3. through ablation studies, the paper demonstrates the contribution of the proposed components to the overall predictive performance.

**Weaknesses:**

The main weaknesses of the article lie in its novelty and results.
* The methods used in the article mostly have already been proposed by others, and the simple stitching together of ideas makes the article lack novelty.
* The experimental results of the article are mostly the second.
* The main experiments in the article are limited. It might be worth considering adding short-term experiments and incorporating new datasets. For example, there are many new datasets available here: https://huggingface.co/datasets/Salesforce/lotsa_data.
* Lack of sensitivity analysis of parameters.

**Questions:**

1. The final projection layer flattens the embeddings of all tokens and maps them to the output, right? So, even though a decoder is used, a separate model still needs to be trained for each length?

---

> ### Author Response · Authors · 2024-11-21
>
> - The experimental results of the article are mostly the second:  The performances of Sentinel are close to the best-published method, i.e. CARD, but behind, the architecture is completely different since Sentinel has a customized attention mechanism designed to leverage the patching structure inherent in time series data and contemporary handling of both temporal and channel dimensions, combining a non-causal encoder for inter-channel dependencies and a causal decoder for temporal dependencies. CARD instead deals with both relationships in the encoder with a totally different attention structure. For this reason, we think the proposed method is interesting also if it does not always outperform CARD.
> - The main experiments in the article are limited: We focused our work on long term foecasting tasks, but we can add the short term forecasting task experiments
> - Lack of sensitivity analysis of parameters: Thank you for your suggestion. We will conduct a sensitivity analysis of the parameters.
> - The final projection layer flattens the embeddings of all tokens and maps them to the output, right? So, even though a decoder is used, does a separate model still need to be trained for each length?: The final projection layer does flatten the embeddings of all tokens and maps them to the output length specified by the prediction length. Yes, you have to train a model for each length. Although a separate model needs to be trained for each length, the use of the decoder is not directly related to enabling dynamic output lengths; rather, the decoder is specifically used to model temporal dependencies.

---

> > ### Comment · Reviewer_NEuV · 2024-11-22
> >
> > My main concerns are as follows, and it seems the authors have not addressed them:
> >
> > 1. Most of the experimental results in this paper perform worse than CARD. In the current "winner-takes-all" era, people tend to adopt the most effective method. This limits the contribution or impact of the paper.
> > 2. In such a scenario, the paper only provides a single major experiment to validate the effectiveness of its approach, which significantly undermines its persuasiveness. I suggest adding new datasets or incorporating other scenarios, such as short-term predictions.
> > 3. Although decoder-only models are not strictly tied to dynamic output lengths, this is a major advantage that distinguishes decoder-only from encoder-only models. In real-world production environments, a single model capable of handling tasks of different lengths is certainly preferred. This diminishes yet another advantage of the paper.
> >
> > Overall, since the authors have not reported new experimental results in the revised paper or elsewhere, and considering the above reasons, I believe the proposed method still has significant room for improvement. Therefore, I will maintain my score.

---

### Official Review · Reviewer_yxmF · 2024-11-04

**Soundness:** 2
**Presentation:** 3
**Contribution:** 2
**Rating:** 5
**Confidence:** 3

**Summary:**

The paper introduces Sentinel, a transformer-based model for time series forecasting that effectively captures dependencies across both temporal and inter-channel dimensions. It further proposes a novel multi-patch attention mechanism to replace the standard multi-head attention in the Transformer architecture. Experimental results suggest the model’s effectiveness in capturing these relationships and forecasting time series.

**Strengths:**

1. The paper is well-written, with a clear presentation of both methodology and results.
2. Experimental results are competitive and close to state-of-the-art (SOTA) performance.
3. The code is made available, which facilitates reproducibility.
4. Targeted Design for Temporal and Inter-Channel Dependency Modeling: The focus on addressing both temporal and inter-channel dependencies in time series data highlights a well-considered model design.

**Weaknesses:**

1. Novelty: The contributions appear to be incremental. The multi-patch attention mechanism provides a marginal improvement over existing architectures, and the modeling of both temporal and inter-channel dimensions in time series is also not a new idea.

2. Justification of Multi-Patch Attention: The rationale behind why multi-patch attention performs better than traditional multi-head attention is not fully explained. An analysis of the root causes for its effectiveness, ideally with theoretical insights or visualizations, would strengthen the contribution.

3. Performance Compared to SOTA: The proposed model does not achieve state-of-the-art performance compared to the provided baselines.

**Questions:**

1. Encoder-Decoder Design Choices: Why does the model use Multi-Patch Channel attention in the encoder and Multi-Patch Time in the decoder? The ablation study in Table 3 provides limited insights—could more analysis clarify the necessity and effectiveness of this design choice?

2. Simultaneous Temporal and Inter-Channel Modeling: Given the importance of both temporal and inter-channel dependencies, could mechanisms be explored that allow these relationships to be learned simultaneously rather than sequentially?

3. Error Bars for Figure 3: Including error bars in Figure 3 would enhance the clarity of the trend and allow for a better understanding of variability in the results.

4. Further Analysis of Multi-Patch Attention: Additional analysis, particularly regarding the effectiveness of multi-patch attention, would provide valuable insights into its utility and potential limitations.

---

> ### Author Response · Authors · 2024-11-21
>
> - Encoder-Decoder Design Choices: We structure our decoder to capture temporal dependencies and our encoder to capture channel-specific dependencies. Our main motivation for this approach is introduced in rows 050-059 and analyzed in the ablation study. In the Decoder, we model temporal dependencies since it is by structure capturing causal relationships.
> During the attention process, the decoder implements a causal mask that allows it to capture only causal relationships.
> This causal structure aligns well with the nature of time series data, where future values should not influence past ones. By positioning temporal attention in the decoder, we can enforce a strict causal order in the forecasting process, enhancing the model’s ability to produce sequential predictions that adhere to real-world temporal constraints. However, the Encoder has no causal mask, allowing it to capture inter-channel dependencies freely across the sequence. Without the constraint of causal masking, the encoder can focus on learning complex relationships between channels across the entire patch, creating a rich contextual representation of channel interactions. In our ablation study, we investigate the impact of specific architectural components by evaluating our model's performance across three distinct datasets with varying feature sizes: Electricity (321 features), Weather (21 features), and ETTh2 (7 features). This choice of datasets allows us to observe the model's behavior under different feature dimensionality, providing insights into how our architecture adapts to diverse data complexities. In Table 3, we show the performance outcomes when different components of our architecture are being modified or removed. Notably, removing the "Channel Encoder", either by adopting a decoder-only structure or replacing it with a traditional "Temporal Encoder", leads to a performances degradation. In particular, the impact is higher in the datasets with more features, highlighting the importance of modeling channel dependencies.
> On the contrary, removing the Temporal Decoder generally negatively affects performance across all datasets. This result underlines the importance of temporal dependencies in our model. It shows that both the channel and temporal encoding mechanisms are essential to achieving robust results across datasets with different feature characteristics.
>
> - Simultaneous Temporal and Inter-Channel Modeling: The architecture of Sentinel already enables both types of relationships to be learned simultaneously. The Encoder learns the channel relationships, while the Decoder learns temporal relationships.
>     These two outputs are integrated through the cross-attention operation in the Decoder, where the self-attention output of the Decoder (capturing temporal patterns) serves as the query, and the Encoder output (capturing channel dependencies) is used as the key and value. This design ensures that the model effectively learns and combines both temporal and inter-channel information, enabling cohesive and accurate predictions.
>
> - Error Bars for Figure 3: Thank you for your suggestion, i can include those bars in the figure

---

> > ### Comment · Reviewer_yxmF · 2024-11-25
> >
> > Thank you for your response. As noted in the previous review, the ablation study in Table 3 remains limited. Simultaneous temporal and inter-channel modeling refers to a module capturing both relationships concurrently—for instance, where both the Encoder and Decoder learn temporal and channel relationships simultaneously. The analysis on this aspect is insufficient. Additionally, the effectiveness of multi-patch attention is not addressed.
> >
> > Since my primary concerns remain unresolved, I will maintain my current score.

---

### Author Response · Authors · 2024-11-21
**Novelty, Contribution and Multi-Patch Motivation**

On Incremental Contributions and Novelty: While it is true that both temporal and inter-channel dimensions have been previously explored, our work introduces a novel integration through the multi-patch attention mechanism with a specialized encoder-decoder architecture. This mechanism reinterprets traditional attention heads by treating patches as independent focus units. Each patch-specific attention acts independently, capturing relationships localized to distinct temporal or channel segments of the sequence. This approach contrasts with standard multi-head attention, which splits attention across heads without exploiting the natural segmentation introduced by patching. Our method leverages the Transformer structure in a structured and efficient manner, combining a non-causal encoder for inter-channel dependencies and a causal decoder for temporal dependencies, ensuring no leakage of future information into past predictions. Furthermore, the specific use of the encoder using its standard non-causal mask to freely capture inter-channel dependencies across the sequence and the decoder with the causal mask to model temporal relationships ensures that future values do not influence past ones. By combining these representations in the cross-attention stage, Sentinel achieves a coherent and efficient integration of multi-dimensional information, which to the best of our knowledge, it has never been proposed before.  To validate the design and effectiveness of the proposed multi-patch attention mechanism, we provide the analysis of an additional synthetic dataset that shows how the attention mechanism works. In this dataset:

- Three dimensions represent linear functions with varying slopes.
- A fourth dimension follows a sinusoidal pattern.

The encoder attention heatmaps derived from this setup will illustrate how different attention patches focus selectively on distinct input segments (temporal and channel-wise), highlighting the disentanglement of dependencies. You can find this visualization in the supplementary material to showcase our mechanism's utility.

---

### Meta-Review · Area_Chair_FCkJ · 2024-12-20

**Metareview:**

This paper proposes an encoder-decoder Transformer model for multivariate time-series forecasting, with a multi-patch attention mechanism to capture both temporal and inter-channel dependencies. The common concerns of all four reviewers is the lack of clear motivation, novelty, and significant performance advantages. The unsolved issues lead to a clear decision of rejection of this paper on its current form.

**Additional Comments On Reviewer Discussion:**

All reviewers acknowledge the rebuttal from the authors but their concerns are no well addressed.

---

### Decision · Program_Chairs · 2025-01-22

Reject